# Characterization of Carbonaceous Deposits on an End-of-Life Engines for Effective Cleaning for Remanufacturing

**Xing Wang** [1,2], **Jianfeng Li** [1,2,*], **Xiujie Jia** [1,2,*], **Mingliang Ma** [1,2] **and Yuan Ren** [1,2]

[1] Key Laboratory of High Efficiency and Clean Mechanical Manufacture, Ministry of Education, School of Mechanical Engineering, Shandong University, Jinan 250061, China; wx1262812261@163.com (X.W.); mingliang_ma@163.com (M.M.); yuan.ren@sdlaser.cn (Y.R.)
[2] National Demonstration Center for Experimental Mechanical Engineering Education, Shandong University, Jinan 250061, China
[*] Correspondence: ljf@sdu.edu.cn (J.L.); xjjia@sdu.edu.cn (X.J.)

**Abstract:** Remanufacturing is one of the most effective strategies to achieve sustainable manufacturing and restore the performance of end-of-life products. However, the lack of an effective cleaning method to clean carbonaceous deposits severely hampers the remanufacturing of end-of-life engines. To explore an appropriate cleaning method, it is necessary to first study the characterization of the carbonaceous deposits. A broad range of analyses including X-ray fluorescence spectrometry, thermogravimetric analysis, [1]H-nuclear magnetic resonance study, X-ray diffraction analysis, and scanning electron microscopy were performed to conduct an in-depth characterization of the carbonaceous deposits. The results showed that a hybrid structure composed of organics and inorganics is the most distinguishing feature of the carbonaceous deposit in end-of-life engines. The inorganics form the skeleton on which organics get attached, thereby resulting in a strong adhesion of the deposit and increasing the difficulty of cleaning. Therefore, a method in which several cleaning forces can be simultaneously applied is more suitable for the present purpose. Molten salt cleaning was chosen to verify the feasibility of this proposal. This method was shown to have the potential to effectively clean the carbonaceous deposit. This finding could contribute towards promoting the effective remanufacturing of end-of-life engines.

**Keywords:** carbonaceous deposits; remanufacturing; cleaning; molten salt

## 1. Introduction

With the excessive consumption of natural resources and the accelerated deterioration of the natural environment, significant attention needs to be directed to sustainable manufacturing [1,2]. As an effective, sustainable manufacturing method, remanufacturing is an important way to reuse end-of-life products and to realize products that may be even better than new ones. This solution can not only alleviate pollution, but also reduce energy and labor consumption during production [3,4]. The remanufacturing steps generally include sorting, inspection, disassembly, cleaning, repairing and reassembly [5,6]. Among these steps, cleaning is regarded as one of the most critical processes because it directly influences and determines whether the subsequent processes can be conducted smoothly [2,7].

It has been reported that about 80% of engine parts can be renewed by remanufacturing. Carbonaceous deposits are one of the most common contaminants deposited on the most important and valuable engine parts, such as valves, pistons, and combustion chambers [8]. Hence, it is necessary to remove these deposits during the end-of-life engine remanufacturing. However, with current cleaning methods, it is difficult to remove these deposits effectively. Therefore, there is an urgent need to devise a suitable and effective cleaning method for carbonaceous deposits. For this purpose, it is essential to study the characterization of the carbonaceous deposits in order to determine the requirements for the cleaning method [9,10].

So far, most of the research on carbonaceous deposits has mainly focused on the following aspects: (1) the factors causing the formation and growth of the deposit [11–13], including temperature, pressure, type of substrate, and the catalyzer; (2) the growth rules of carbonaceous deposits [14,15]; (3) the effect of carbonaceous deposits on the engine performance [13,16]; and (4) the chemical reactions occurring during the formation of carbonaceous deposits [12,17]. Characterization analysis and related research on carbonaceous deposits lacking; studies have mostly dealt with the characterization of composition [18] and surface morphologies [19]. Haji-Sulaiman et al. classified carbonaceous deposits into "dry deposits" and "oily deposits" according to their morphology and content of volatile organic matter [16]. Diaby et al. analyzed the aromatic components formed in the grooves of pistons of diesel engines [14]. Smith et al. found some inorganic matter in heavy-duty diesel engine piston deposits and studied their distribution [20]. The deposits investigated in these studies were all formed in simulated experimental platforms within 500 h. In contrast, the formation of deposits in end-of-life engines accompanies the entire service life of parts, which is generally much more than three years. It can be observed that the formation time of the deposits in the end-of-life engines is much longer than the deposits in the above studies. As the composition and structure of the carbonaceous deposits will vary with their growth processes [21–23], it is necessary to study the carbonaceous deposits, particularly in end-of-life engines.

Moreover, there is a lack of studies about the structure of the deposits, which is a vital factor affecting the grading of contaminant removal—that is, easy or difficult [24]. Therefore, the study of the carbonaceous deposit in an end-of-life engine was conducted regarding their composition and structure. Specifically, the aim of this study was to investigate the reason for the difficulty in removing carbonaceous deposits and to provide guidance for exploring suitable and effective cleaning methods.

## 2. Experimental

### 2.1. Sample

The samples studied were carbonaceous deposits covering the surface of a valve stem in a WD615.47 diesel engine, which was an end-of-life engine. The model of the valve stem was VG15060051001 and it had been used for at least five years.

The carbonaceous deposits analyzed in this study were in two forms: powder and bulk states. The powder sample was the contaminant layer stripped from the surface of the polluted valve stem and then ground in a mortar. A bulk sample of size $10 \times 10 \times 10$ mm$^3$ was cut by wire-electrode cutting.

### 2.2. Characterization Analysis

The surface and the cross-sectional morphology of the sample were observed by scanning electron microscopy (SEM; FEI QUANTA FEG 250, FEI, Hillsboro, OR, USA). The elemental composition was analyzed via energy dispersive X-ray spectroscopy (EDS; Oxford Instruments, England), and X-ray fluorescence (XRF; EDX1400, Shimadzu, Japan). The composition analysis was conducted by $^1$H-nuclear magnetic resonance study ($^1$H-NMR; AVANCE-HD NMR, Bruker, Germany), and X-ray diffraction analysis (XRD; D8 Advance diffractometer system, Bruker, Germany). The thermal behavior of the samples was studied by thermogravimetric analysis (TGA) with a thermobalance (METTER-TOLEDO, Zurich, Switzerland).

SEM: The sample used for SEM analysis was in the bulk state. The sample was embedded in inlaid powders using a mosaic machine for sectional micrograph observation.

FT-IR: Powdered carbonaceous deposit (approximately 1 mg) was mixed into KBr (approximately 200 mg) and ground to 2 μm. The mixed powder was then compacted into a disc (with a diameter of 13 mm and a thickness of 1 mm) by a press machine at 20–26 MPa for 2 min. The spectra were recorded in the range of 4000–500 cm$^{-1}$ at a resolution of 2 cm$^{-1}$. Before detection, pure ground KBr was used to obtain a reference spectrum.

XRD: The powdered samples were first ground to approximately 50 μm. The microparticles were then mounted on non-diffracting silica substrates and analyzed with Cu Kα radiation (30 kV, 30 mA). The scattered X-ray intensities were collected with a scan speed of 2°/min over an angular range of 10°–70° (2θ).

TGA: The thermal behavior of the sample was analyzed in nitrogen ($N_2$) and oxygen ($O_2$). About 40 mg of the powdered carbonaceous deposit was analyzed in an alumina pan. The temperature ranged from 0 °C to 800 °C at a heating rate of 10 °C/min.

XRF: The powdered carbonaceous deposit sample was dried in a drying oven for half an hour The powders mixed with the dried sample and binder (starch) were then ground using a motor and then put into an aluminum cup. The powder mixture was then pressed with a tablet press. The prepared sheet sample was then ready to be analyzed.

$^1$H-NMR: Tetramethylsilane (($CH_3)_4$Si, abbreviated as TMS) was used as the standard substance for detection. The powdered carbonaceous deposit and TMS were dissolved together in the solvent (the solvent was deuterated chloroform ($CDCl_3$)).

Three parallel experiments were performed with the carbonaceous deposit samples taken from three valve stems of the valve system. The experiment was designed to avoid one-time occasionality in the analysis and ensure the validity of the test.

### 2.3. Molten Salt Cleaning Procedure

The molten salts were composed of 35 wt% NaOH and 75 wt% KOH. The reagents used were all analytically pure (Sinopharm, Beijing, China). For each run, 100 g of the mixed salt was used. The mixed salts were placed in a stainless-steel container and heated in a universal electric furnace (Lichen 180, Shanghai, China) with the cleaning temperature maintained at 250 °C (which refers to the surface temperature of the molten salt system). The cleaning process lasted for 2 min, after which the sample was removed and cooled down in the air to 60 °C. There was some molten salt that remained, which crystallized on the sample's surface after cleaning. The cleaned sample was immersed in a water bath to remove the remaining salt. The samples used in the cleaning process were a quarter of the valve stem. Ten samples from different valve stems were selected for the cleaning experiment.

## 3. Results and Discussion

### 3.1. Composition

The elemental contents detected by XRF are shown in Table 1. The carbonaceous deposit comprised 42% metal elements (calcium, iron, and zinc) and 51% non-metallic elements (carbon, oxygen, silicon, phosphorus, and sulfur). Carbon (C) and metallic elements (Ca, Fe, and Zn) were the main and vital elements in the carbonaceous deposit. The proportion of metallic elements in the deposit was much higher than what has previously been reported in the literature [18,25]. This result indicates the high inorganic composition of the carbonaceous deposits on the parts of an end-of-life engine. Carbon is the most abundant element in the carbonaceous deposit, and it may exist in two forms: in the crystalline state as carbon, and as amorphous carbon in the organic matter [18]. The TGA results are shown in Figures 1 and 2 and can be used to further identify the state in which C exists in the deposit.

The curve of the pyrolysis of carbonaceous deposit can be divided into three regions according to the shape of the curve: region I, with temperatures below 200 °C; region II, with temperatures of 200–470 °C; and region III, with temperatures of 480–800 °C. In regions I and II, the shape of the curve is similar under different conditions, which means that pyrolysis was the main reaction that occurred below 480 °C, as shown in Figures 1 and 2. In region III, the rate of matter loss showed by the derivative thermogravimetric (DTG) curve in Figure 2 is much higher than that in Figure 1, which indicates some components can be oxidized to gas in this region.

**Table 1.** Major elements in the carbonaceous deposit.

| Element | Mass Fraction (%) |
|---|---|
| C | 24.3559 |
| O | 10.4700 |
| S | 8.7396 |
| Ca | 18.4773 |
| Fe | 12.2681 |
| Zn | 12.1216 |
| Si | 4.2892 |
| P | 3.1958 |
| Al | 2.3089 |
| Other Metals | 3.7734 |

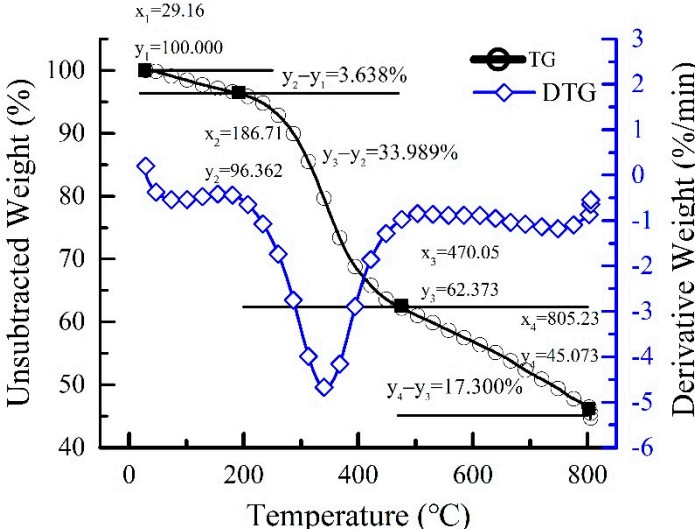

**Figure 1.** Thermogravimetric analysis (TGA) results in nitrogen environment.

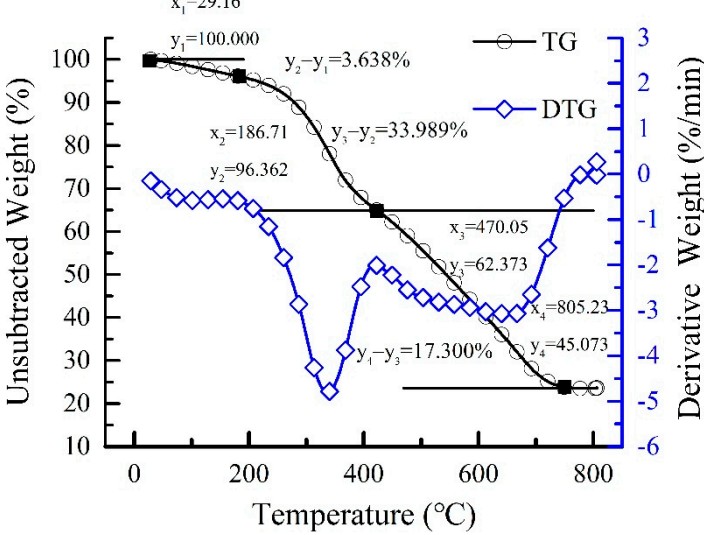

**Figure 2.** TGA results in oxygen environment.

The similar matter loss under two different atmospheres in the first two regions can be interpreted as the evaporation of light compounds, including water and organics [22]. They comprised about 35% of the content in the carbonaceous deposit. The matter lost in region II mainly referred to the higher-molecular-weight organics, which suggests that the

organic components in the deposit were macromolecular. There was an obvious difference in region III when analyzed in a different atmosphere, where about 17% of the matter was lost in nitrogen and about 41% in oxygen. This result illustrates that about 24% of the contents were oxidized and lost at this temperature. According to the elemental content shown in Table 1, this can only be attributed to the oxidization of crystalline carbon. Further, slow and gentle pyrolysis occurred in region III in nitrogen, representing the pyrolysis of polymers with very high molecular weights. Therefore, the crystalline carbon oxidized in region III may arise from the pyrolysis of the polymers or from the original content of the deposit.

Note that about 23.5% of the composition remained after region III in Figure 2, which indicates the existence of inorganic matter. Therefore, the carbonaceous deposit in the end-of-life engine was mainly composed of inorganic and high-molecular-weight organic matter.

### 3.2. Organic and Inorganic Components in the Deposit

In our previous study [26], the organic component of this deposit had been analyzed by Fourier transform–infrared (FT-IR) spectrometry. The results showed the strong absorption of methyl (-CH$_3$-) and methylene (-CH$_2$-) at 2800 cm$^{-1}$ to 3000 cm$^{-1}$ and 1300 cm$^{-1}$ to 1500 cm$^{-1}$, respectively. The $^1$H-NMR spectrum shown in Figure 3 was used to furtherly analyze the structure of the organic components by detecting the position of the hydrogen.

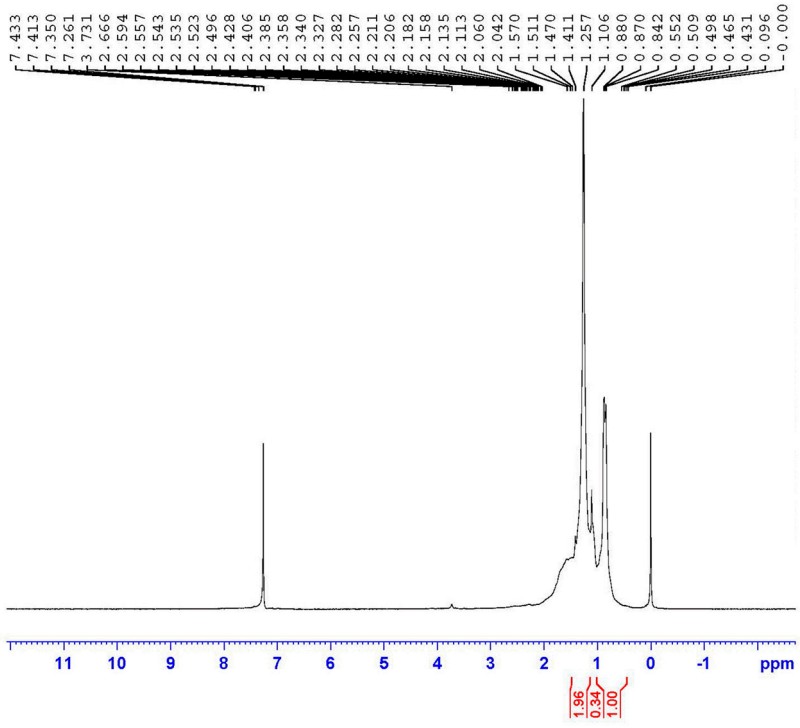

**Figure 3.** $^1$H−nuclear magnetic resonance (NMR) spectroscopy of the carbonaceous deposit.

There are three groups of chemical shift peaks shown in Figure 3. The shift values δ H = 0 ppm and δ H = 7.26 ppm are the basic material (TMS) and solvent (CHCl$_3$). The chemical shift at 0.8~1.3 ppm represents the elemental hydrogen on saturated carbon. The main peaks of δ (ppm) H(a) = 1.25, δ H(b) = 0.88, δ H(c) = 1.11, and δ H(d) = 1.50 represent CH$_2$-R, CH$_3$-R, R-CH$_2$-R (cyclic), and CH-R, respectively. This indicates that the hydrogen in the organic matter originated mainly from alkanes. Based on the features shown in Figures 1 and 2, the organic matter in this carbonaceous deposit is considered to be a mixture of linear alkanes, naphthenic compounds, and their polymers.

Figure 4 further provides more insight in the analysis of the inorganic matter in the deposit. It shows an obvious background at 10°–30°, caused by amorphous organic

matter. The characterization of the complex composition of the deposit is also reflected by the disordered and irregular shapes of the peaks. The sharp peaks at 25.5°, 26.7°, and 29.5° (2θ) are assigned to anhydrite ($CaSO_4$), graphite (C), and calcium pyrophosphate ($Ca_2P_2O_7$), respectively, based on the elements contained in the carbonaceous deposit. It can be found that the inorganic components were mainly additives in fuel or lubrication oil and crystalline carbon, which reflects the origin of the inorganic matter contained in the carbonaceous deposit. This also suggests that there was some crystalline carbon (C) contained in the carbonaceous deposit originally.

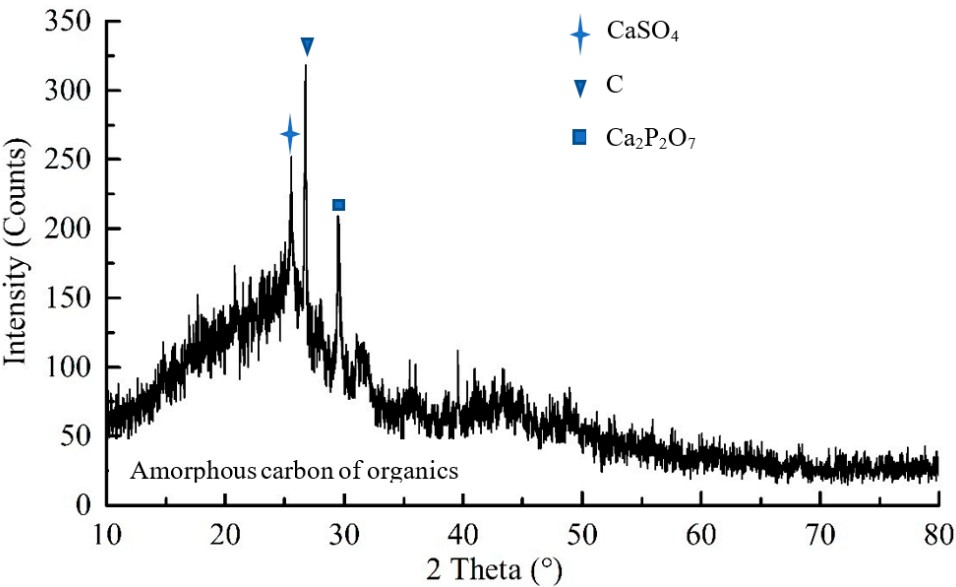

**Figure 4.** X-ray diffraction (XRD) analysis result of carbonaceous deposit.

Therefore, the composition of the carbonaceous deposit in the end-of-life engine parts was a mixture of macromolecular alkanes, carbon, and calcium salts. The organic matter in the carbonaceous deposit was also mainly the product of the combustion of fuel and lubrication oil [20,23]. The composition of the carbonaceous deposit was verified to originate from the fuel and lubrication oil. Therefore, the components in the carbonaceous deposit varied with the fuel or lubricant oil used in the engines as well as the working condition. This suggests that the specific composition of the carbonaceous deposit was different in different parts of the engine.

### 3.3. Structure Analysis

The high temperature in the engine results in most of the volatile products being exhausted, leaving the alkanes and their polymers behind. This is because of the relatively high thermal stability of the saturated organics. However, the temperature in the engine was still too high to retain the alkanes and their polymers. This is the result of the special structure composed of organic and inorganic particles, which will be shown and explained in the following structure analysis.

The cross-sectional micromorphology and element content variation from the interior to the surface as detected by the EDS surface scanning of the carbonaceous deposit is shown in Figure 5. The figure shows the synchronous content variation of iron and oxygen in the inner layer, which indicates that the iron oxide (generally called the denaturation layer) was produced before the contaminant was deposited [27]. It appears that the denaturation layer, which acts as a layer for the attachment of the carbonaceous deposit, plays a vital role in the formation of the carbonaceous deposit. As the carbonaceous deposit attached to the denaturation layer, the carbon content did not clearly increase synchronously. However, at the surface, the carbon content began increasing, indicating that the organics mainly covered the surface of the deposit and were not evenly distributed in the whole deposit

layer. Instead, the inorganic components showed relatively higher content in the inner layer. The micromorphology observed at high magnification in Figure 6 shows that the inner deposit comprised both organic and inorganic matter. This is because the high temperature in the engine is not conducive to the deposition of organic matter, and only a small amount of organic matter, protected by inorganic particles, remained. Therefore, the inner part of the deposit is very stable and cannot be easily removed.

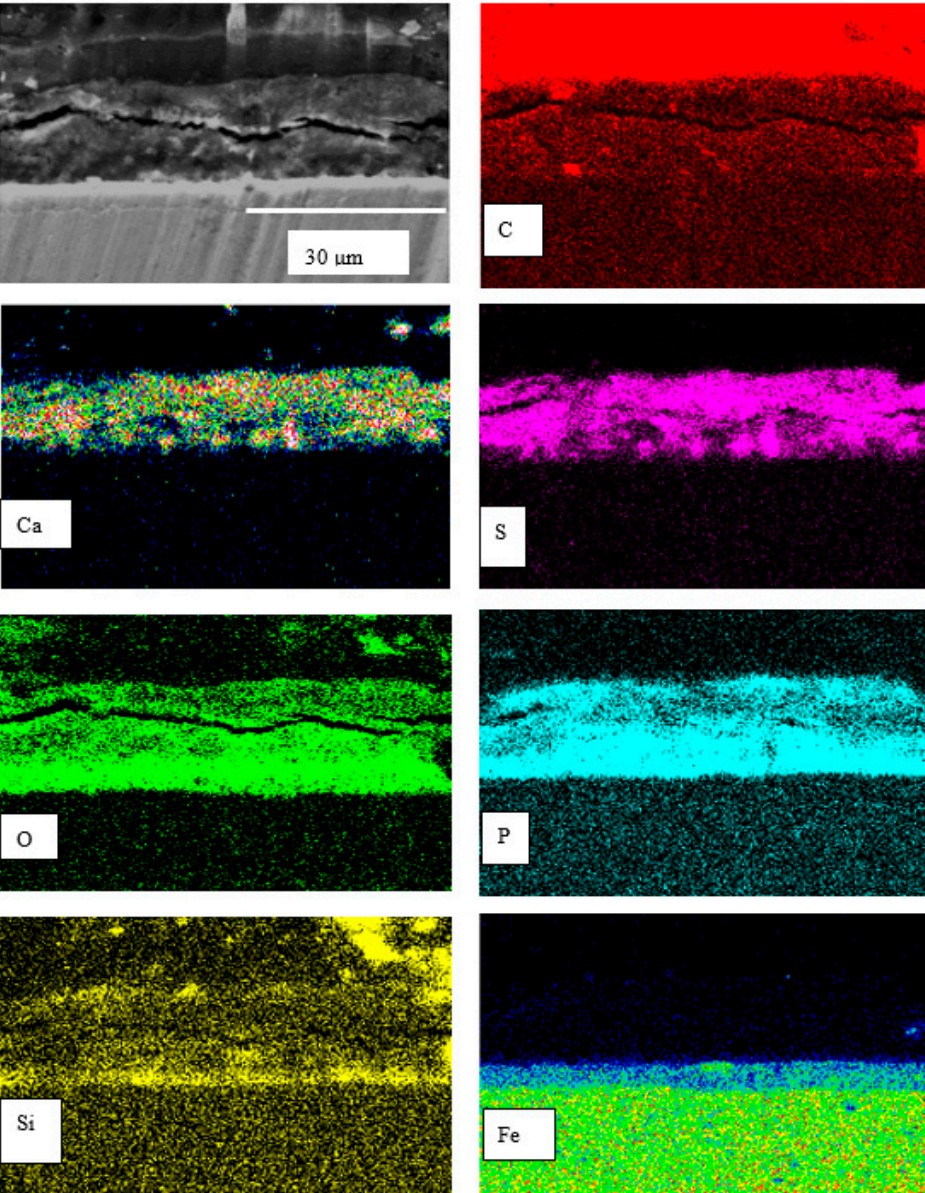

**Figure 5.** Surface scanning results for the cross section of the carbonaceous deposit.

Figure 7 shows the morphology of the carbonaceous deposit at different magnifications. The deposit shown in the images can be classified into two groups: particles (inorganics) and other matter (organics). The inorganics are surrounded by organic matter. The particle size of the inorganics was about 1–30 µm and was distributed unevenly. The organic matter consisted of nanoparticles smaller than 50 nm which could be observed under 80,000× magnification. The inorganic particles, acting as a surface for attachment, extended the surface area available for depositing the organics and prevented the evaporation of inner organics owing to their low thermal conductivity. Other researchers also found that the inorganics can enhance the stability of the organics [24], which retains the alkanes at a

relatively high temperature. Further, the organics bond the inorganic particles together, enhancing the stickiness of the surface to enable absorbing more inorganic particles for dense deposition. There is interdependence and an interactive relation between the organic components and inorganic components in the carbonaceous deposit, which is the foundation for forming the hybrid structure of the deposit. Hence, the interaction between the organics and inorganics promotes the growth of the carbonaceous deposit and makes them more stable and more difficult to remove than contaminants with a simple structure.

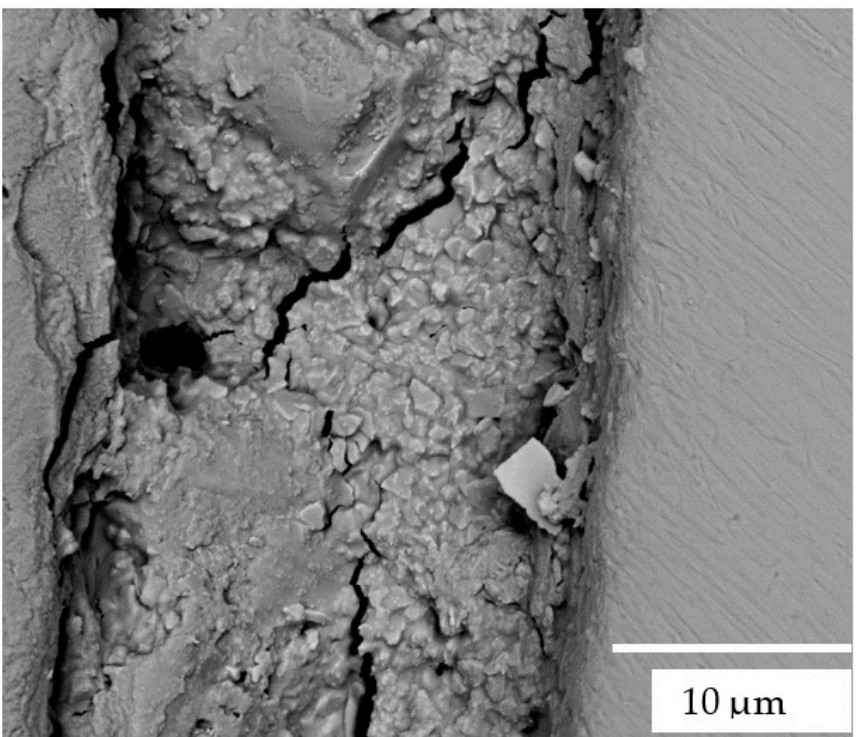

**Figure 6.** Micromorphology of the inner layer in the carbonaceous deposit observed at high magnification.

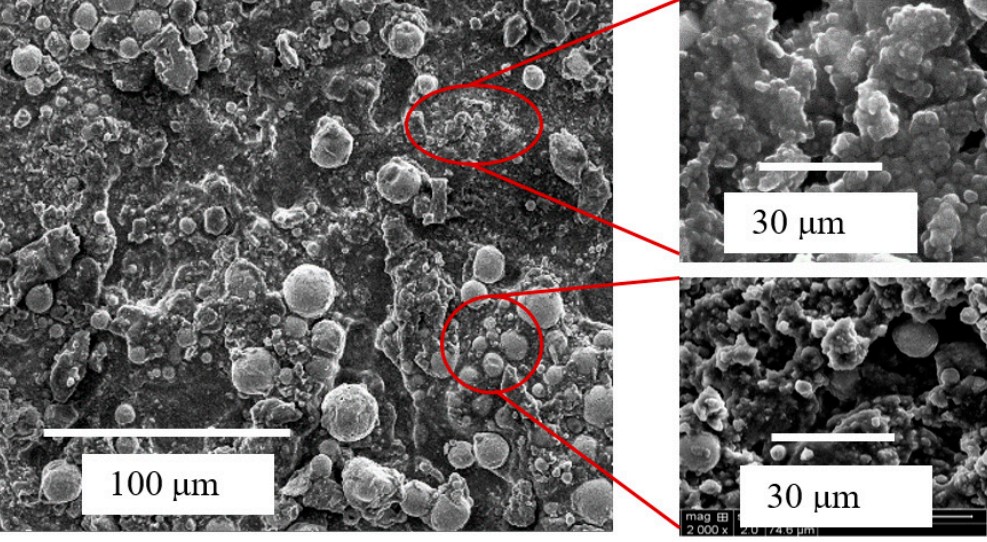

**Figure 7.** Scanning electron microscopy (SEM) images from the surface of the carbonaceous deposit at different magnifications.

By providing a surface for attachment, the inorganics not only enhance the stability of the organics but also form a skeleton to ensure the growth of the carbonaceous deposit, as the schematic diagram shows in Figure 8. The organic and inorganic contents may differ depending on the fuel or lubricant oil and the working condition. However, the hybrid structure composed of organics and inorganics is the most distinguishing feature of all carbonaceous deposits in end-of-life engines, and it plays a key role in hampering efforts for their removal. This result emphasizes the key role played by inorganic particles in the formation and growth of carbonaceous deposits.

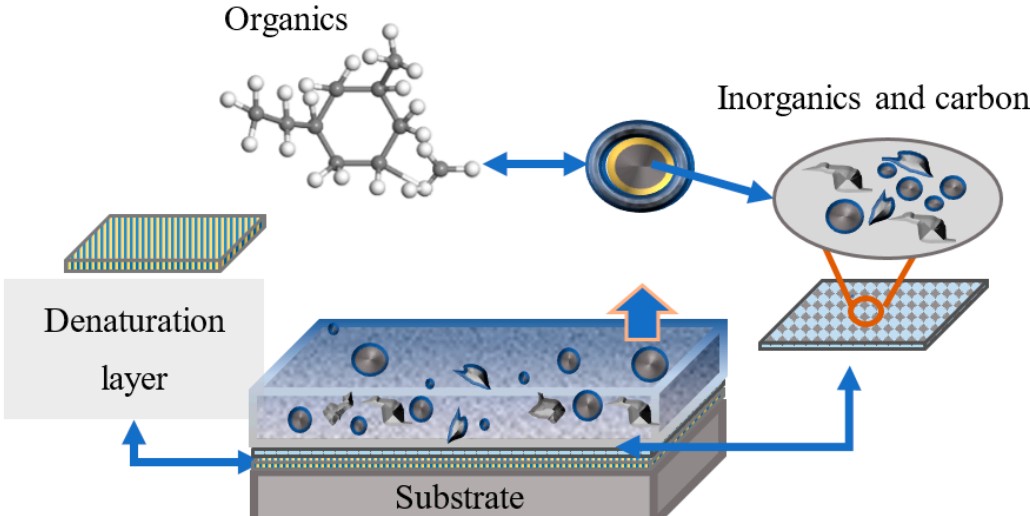

**Figure 8.** Schematic diagram of the structure of the carbonaceous deposit.

Generally, cleaning methods for organic contaminants are not effective against inorganic particles. Therefore, the cleaning method should be able to clean mixed contaminants. Besides, the inorganic matter is mainly distributed in the inner layer, suggesting that the organics should be cleaned in the first step. These concerns imply that for the removal of a carbonaceous deposit, compound methods or a method with compound effects are required.

*3.4. Cleaning Effect of Molten Salt on the Carbonaceous Deposit*

As a high-efficiency cleaning method, molten salt cleaning provides a compound effect against contaminants and is a robust cleaning method for contaminants with complex composition. The compound cleaning effect of this method on the carbonaceous deposit was tested in this study.

Table 2 shows the pictures of samples before and after molten-salt cleaning. The pictures in the third row show that the samples could be cleaned by a 2 min molten salt treatment. Hardly any contaminant can be observed by the naked eye after the cleaning. For samples (a) and (b), the color of the substrate was exposed after cleaning. However, a colorful surface is seen in the case of sample (c), and it represents the oxidation of the substrate. This difference in the color of the substrate is related to the different elements on the surface of the substrate. The results indicate that the method is effective for carbonaceous deposit removal.

Figure 9 shows the surface micromorphology of the sample after molten-salt cleaning at different magnifications. Figure 9a shows that the surface structure of the substrate could be clearly observed after cleaning. With further magnification to 2000×, it can be seen that some bright particles covered the surface irregularly, as shown in Figure 9b. The organic matter could no longer be observed, even at 5000× magnification (Figure 9c). The components at point (i) and point (ii) are mainly the oxidation products of metals. The results demonstrate that no carbonaceous deposit remained on the surface after molten-salt

cleaning. The organic matter and inorganic particles were removed effectively by molten-salt cleaning, indicating that this is an effective method to remove carbonaceous deposits. It also verifies the above analysis and the feasibility of the proposed cleaning method for removing the carbonaceous deposits on end-of-life engines.

**Table 2.** Pictures of the sample before and after molten-salt cleaning.

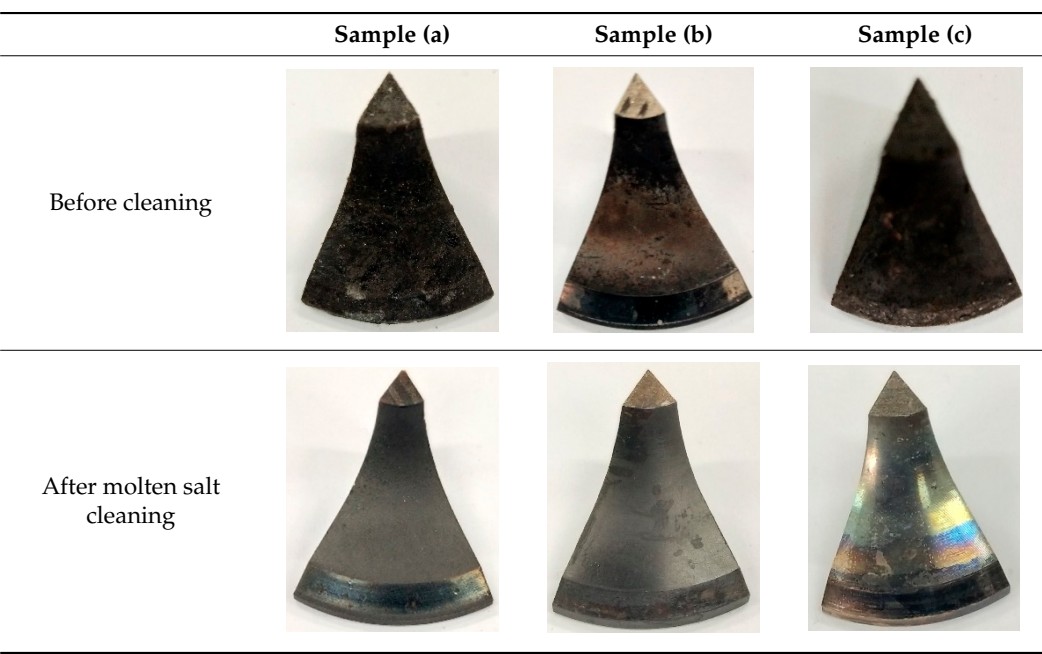

| | Sample (a) | Sample (b) | Sample (c) |
|---|---|---|---|
| Before cleaning | | | |
| After molten salt cleaning | | | |

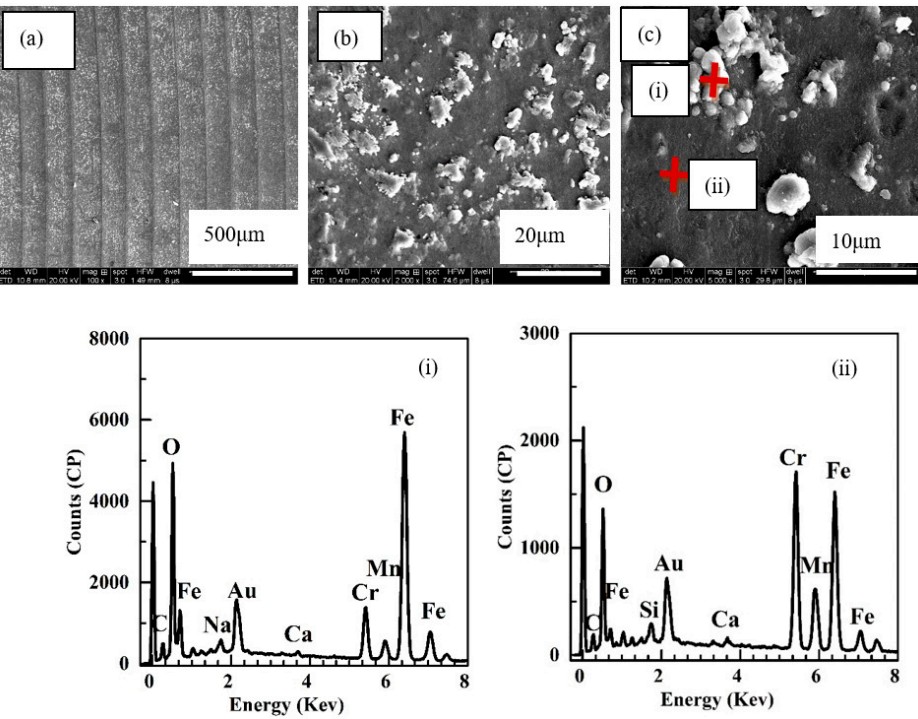

**Figure 9.** SEM images from the surface of the carbonaceous deposit after molten-salt cleaning. (**a**) With $100\times$ magnification, (**b**) with $2000\times$ magnification, and (**c**) with $5000\times$ magnification, (i) elemental composition of point (i), (ii) elemental composition of point (ii).

## 4. Conclusions

To explore a suitable and effective cleaning method for removal of carbonaceous deposits on end-of-life engines, a characterization analysis was performed to study the requirements for the cleaning method in this research. The XRF and TGA found that the carbonaceous deposit was composed of both organic and inorganic matter. The composition analysis conducted by [1]H-NMR and XRD revealed the organic and inorganic composition and their origin. The structure analysis by SEM finally clarified the reasons for the challenges in removing the carbonaceous deposit on the end-of-life engine.

The contributions of this work are as follows:

1.  The important roles of organics and inorganics in the carbonaceous deposits in end-of-life engines have been confirmed. The interlayer of organic and inorganic matter enhances the stability of the deposits. The most distinguishing feature of carbonaceous deposits on end-of-life engines is the hybrid structure composed of organics and inorganics.
2.  The organic matter was mainly comprised of alkanes and their polymers. The inorganic matter mainly comprised inorganic salts and crystalline carbon. It all originated from fuel or lubricant oil.
3.  There is a need for a compound cleaning method or a cleaning method with compound effects to remove carbonaceous deposits on end-of-life engines. The effect of molten-salt cleaning on this deposit were tested, which verified that the proposed cleaning method is feasible and effective. However, further cleaning is needed for the denaturation layer. Therefore, further research in this direction is needed to explore a more effective compound cleaning method for this hybrid structure.

This study can inform the exploration of more effective methods for the removal of carbonaceous deposits on end-of-life engines. The results of this work could help to accelerate the reuse of end-of-life parts with carbonaceous deposits. It could not only promote the progress of sustainable manufacturing, but could also help to make cleaning technology more environmentally friendly. However, the samples analyzed in the study were extremely limited. More detailed characterization studies of carbonaceous deposits should be conducted with more samples, exploring the relative content of organic and inorganic matter in these deposits.

**Author Contributions:** Conceptualization, X.W. and J.L.; Methodology, X.W. and Y.R.; Validation, X.W. and M.M.; Writing—Original Draft Preparation, X.W. and X.J.; Writing—Review and Editing, J.L.; Visualization, X.W. and M.M.; Supervision, J.L.; Funding Acquisition, J.L. All authors have read and agreed to the published version of the manuscript.

**Funding:** This research was funded by Major Project on Science and Technology Innovation of Shandong Province, grant number 2018CXGC0807 and National Natural Science Foundation of China, grant number NSFC51875324.

**Institutional Review Board Statement:** Not applicable.

**Informed Consent Statement:** Not applicable.

**Data Availability Statement:** No new data were created or analyzed in this study. Data sharing is not applicable to this article.

**Conflicts of Interest:** The authors declare no conflict of interest.

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
