# Peer review of "Characterization of Carbonaceous Deposits on an End-of-Life Engines for Effective Cleaning for Remanufacturing"

_sustainability, doi:10.3390/su13020950_

Round 1

Reviewer 1 Report

The authors Wang et al. have studied the characterization of carbonaceous deposit on spent diesel engines for their effective cleaning in remanufacturing process. The work is well suited to this journal and the study seems to have a direct implication on the remanufacturing industry.

  1. The language of the manuscript is poor and needs a thorough revision including some typos.
  2. When analyzing any carbonaceous material, the analysis of loss of ignition and volatile carbon is very primary information to have. I suggest to include this in Table 1.
  3. All the figures are blurred and looking pity. Hence, they must be supplied in high resolution.
  4. The reviewer would like to suggest to supply the FT-IR analysis of the studied carbonaceous material as well.
  5. The authors need to provide more information on the denaturation layer after the molten salt cleaning.

Reviewer 2 Report

This manuscript reports the characterization analysis of carbonaceous deposit on end-of-life engines and also effective cleaning method for remanufacturing. However, the authors did not convey the main aim/objectives of this work clearly. The authors have used molten salt cleaning method to remove the carbonaceous deposit in this work, but have mentioned in the introduction that this method faces the problem of low effectiveness, high energy consuming and high economic cost, etc., which is contradictory and confusing. Although the characterisation analysis of the deposit on engines seems to be acceptable, this work may not be published due to incompleteness and lack of complete story and aims. I would suggest the authors to rewrite the paper with clear introduction as why the authors are carrying out this work i.e. main aims/objectives, what they intend to achieve, methods adopted and the reason for choosing that particular method, results and discussion and finally conclusions.

Reviewer 3 Report

In this paper, the authors have characterized a carbonaceous deposit on end-of-life engines using X-ray fluorescence spectrometer, thermogravimetric analysis, H-nuclear magnetic resonance and scanning electronic microscopy. The research study is limited and I have strong reservation on the results quality. Overall, I don’t see the potentiality of this study.

The introduction is not informative. I suggest to improve the introduction focusing on the previous researches conducted specifically on the characterization of component and microstructure for carbonaceous deposit on end-of-life engines. Moreover, I suggest to clarify the implications of your study. Consider extending the dataset to more samples to prove unambiguously that your results have potential in the remanufacturing progress of end-of-life engines.

The methods have been applied correctly and the multi-methodological approach would provide a good viewpoint on the characterization of the sample under study. However, the experimental protocols are not described in sufficient detail that would allow others to repeat the experiments. The data set is very limited and thus not straightforward to support the hypothesis of the study. Most of the results are poorly discussed not adding contributions to the body of knowledge. For these reasons, I do not recommend the publication in the present form.

Round 2

Reviewer 1 Report

The manuscript has reached up an improved level.

Reviewer 2 Report

The authors have made significant changes to the manuscript. However, the English and the writing style has to be checked or worked up on.

Reviewer 3 Report

The authors took my proposed changes into account when producing the revised version. Data are well discussed and all recommendations have been addressed in the revised version. As for these reason, I suggest the publication of the paper with minor revision. Please consider the comments below to revise your manuscript:

  • Please consider to use the journal template to prepare the manuscript
  • Abstract: Please change “this finding will promote” with “this finding could contribute to promote”
  • Introduction: “However, the deposit studied in these researches had all formed in a simulated experimental platform within 500 h, which is much shorter”. How much shorter? Please be more specific.
  • 2. Characterization analysis: Please change “on non-diffracting silica substrates and detected with” with “on non-diffracting silica substrates and analyzed with”
  • 2 Organic and inorganic components in the deposit: please change “Therefore, the kind of ingredient in the carbonaceous deposit will” with “Therefore, the components in the carbonaceous deposit will”.
